# Trace Elements in Beef Cattle: A Review of the Scientific Approach from One Health Perspective

**DOI:** 10.3390/ani12172254

**Published:** 2022-08-31

**Authors:** Fernando Luiz Silva, Ernandes Sobreira Oliveira-Júnior, Marcus Henrique Martins e Silva, Marta López-Alonso, Maria Aparecida Pereira Pierangeli

**Affiliations:** 1Department of Education, Federal Institute of Mato Grosso, Alta Floresta 78580-000, MT, Brazil; 2Center of Ethnoecology, Limnology and Biodiversity, Laboratory of Ichthyology of the Pantanal North, University of Mato Grosso State, Postgraduate Program of Environmental Science, Cáceres 78200-000, MT, Brazil; 3Geoprocessing Nucleus, Federal Institute of Mato Grosso, Alta Floresta 78580-000, MT, Brazil; 4Department of Animal Pathology, University of Santiago de Compostela, 27002 Lugo, Spain; 5Department of Animal Science, Postgraduate Program of Environmental Science, University of Mato Grosso State, Pontes e Lacerda 78250-000, MT, Brazil

**Keywords:** environment, farm, food safety, livestock, sustainability

## Abstract

**Simple Summary:**

This review seeks to understand how scientific research on trace elements in beef cattle has been developed and how the interaction of this theme with topics related to animal, environmental and human health has been established. Given the duality of many of the trace elements, being known as nutrients in small amounts or toxic when they exceed small concentrations, we brought the One Health perspective to analyse how researchers approach this research theme. In this work, we propose a path through scientific production to promote innovation, sustainability of animal production, food safety and human health.

**Abstract:**

The objective was to investigate the context, approach and research topics present in the papers that analysed trace elements in beef cattle to identify gaps and scientific perspectives for the sustainable management of trace elements in livestock. The main research groups came from the United States, Spain, Japan, Brazil, India and Slovakia, which represented 31% of the papers produced. Only 37% of studies addressed aspects that integrated animal, environmental and human health. The reviewed papers concerned 56 elements and 15 bovine tissues (Cu, Zn, Pb, liver, muscle and kidney highlighted). The main gaps were (1) lack of research in developing countries, (2) the need to understand the impact of different environmental issues and their relationship to the conditions in which animals are raised, and (3) the need to understand the role of many trace elements in animal nutrition and their relationship to environmental and human health. Finally, we highlight possible ways to expand knowledge and provide innovations for broad emerging issues, primarily through expanding collaborative research networks. In this context, we suggest the adoption of the One Health approach for planning further research on trace elements in livestock. Moreover, the One Health approach should also be considered for managers and politicians for a sustainable environmental care and food safety.

## 1. Introduction

Trace elements occur in the environment at concentrations typically below 100 mg kg^−1^ [1]. They are described as “trace” because they could not be quantified by the first analytical methods available [2]. Trace elements can be nutritional elements, such as Cu, Fe and Zn, or potentially toxic elements, such as Cd, Hg and Pb [3]. Although they occur naturally at low concentrations, trace elements are of great importance. For example, the presence of small variations in the concentrations of trace elements in cattle feed can cause a substantial reduction in performance and production or can result in deficiency or toxicity, even leading to death of the animals [4].

The concentrations of trace elements in the environment and foods vary widely. Thus, in the context of the global environmental change that is occurring in the so-called Anthropocene era [5] and considering that the concentrations of trace elements are influenced by a series of factors of natural or anthropogenic origin, such as geological and edaphic factors or agricultural and industrial emissions [6,7], there is an ever-present risk that changes will cause ecological imbalance in the most diverse environments. 

Within the agriculture sector, mineral supplementation in animal production is of great importance. It is well known that an inadequate (generally deficient) supply of essential trace elements leads to poor animal condition, adversely affecting immunity and reproduction [8]. Historically, extensive areas of trace element deficiency have been identified worldwide, and a vast and comprehensive amount of research (for the time) was conducted some decades ago to establish trace element requirements and the metabolic role of trace elements [2,9]. In this context, trace element supplementation, mainly as mineral mixtures added to animal concentrate feed, was routinely established as “insurance” for the herd; this was possible because most trace elements have wide safety margins, and the cost of the minerals was offset by the benefits gained [4]. This led to the release of large amounts of trace elements to the environment with nowadays well-known diverse negative consequences, such as toxic effects on plants, microorganisms and animals [10]. Furthermore, to a lesser extent, animals reared in polluted areas (mainly due to industry and mining) can be exposed to high levels of toxic elements and develop signs of acute or chronic toxicity [11,12].

Trace element exposure in livestock also has important consequences for humans. It is well known that meat and meat products (mainly offal) are some of the main contributors of trace elements, including toxic elements, and the concentrations of these elements in meat are directly related to those in animal feed [13]. High levels of toxic elements, often exceeding regulatory limits, are found in meat products in polluted areas [14]. Moreover, excessive supplementation of trace elements, particularly for elements (such as Co, I and Se) for which intestinal homeostatic mechanisms of absorption do not exist, leads to over-enriched animal products [4]. Traces of toxic elements present in mineral supplements and feed can also accumulate in animals and increase levels of these elements in meat products [15]. In addition, interaction with other elements may occur and reduce the uptake of essential elements (e.g., Cd-Cu and Cd-Zn) [16].

The triple challenge of trace element exposure in livestock—to optimize animal health and productivity while ensuring environmental sustainability and consumer/food safety—leads to the need to consider trace element exposure, in particular trace element supplementation, from a holistic perspective, which is nowadays referred to as One Health. The One Health approach aims to ensure the well-being of people, animals and the environment through collaborative problem solving, integrating these highly interconnected components and enabling change towards better public health outcomes [17]. Although the One Health concept was originally born from a zoonotic perspective, in recent years the concept has expanded from the medical and veterinary sciences to include a rapidly growing range of synergistic disciplines, including food safety, public health, health economics, ecosystem health, social science and animal health [18]. It is now recognized that environmental factors, including chemical contaminants in animals and animal products, residues of additives and veterinary drugs and phytosanitary products play a significant role in human, animal and plant health. Thus, the need for a One Health approach in nutrition/animal production, research and policy is a priority.

During the last few decades, a large body of research has addressed trace elements in farm animals. Most of this research has focused on using trace elements, even at concentrations above physiological requirements, to improve animal performance. However, many challenges remain in understanding the dynamics of these elements from the animal/farm to the consumer to produce profitable but sustainable farms and thus accurately determining the possible short, medium and long-term impacts. Some aspects of animal production, such as breed and physiological state [19,20], antagonism between elements [21], productive practices adopted in livestock [22] and air quality and environmental quality in general [23,24,25], can have significant impacts on the concentrations and accumulation of these elements in animal tissues. The precise management of these elements in animal production is not an easy task, particularly in extensive production systems, due to the existence of a wide variety of environments, diversity of management practices, complex interactions between elements and biochemical processes and the occurrence of natural and anthropogenic changes in the environment [7,26,27].

The aim of this review was to collect and globally summarize the research carried out in the last few decades concerning trace elements in beef cattle and analyse the approach of researchers through bibliometric information (years, countries, authorship and journals) and topics (trace elements, bovine tissues, analytical techniques and animal, environment and human health) presented in their respective papers. Beef cattle were selected from among other livestock species as they are possibly the type of livestock that best exemplifies the triple animal–environment–consumer interaction and in which the One Health approach to trace element exposure would thus be of greatest benefit. We hope that this paper will identify the main gaps and prospects for advancing knowledge about trace elements in beef cattle production. This would facilitate future collaborations and enable responsible use of trace elements in animal farming while protecting animals, consumers and the environment.

## 2. Materials and Methods

We carried out a search in the Scopus, Web of Science and PubMed scientific databases. To retrieve the papers that analysed trace elements in beef cattle tissues, advanced search strings were created using five sets of keywords in English, which appeared in titles and abstracts of papers published between 2000 and 2022. The search interval from 2000 onwards was defined to cover the United Nations Millennium Summit initiative held that year when the Millennium Development Goals were created and which in 2015 culminated in the proposal of the 17 Sustainable Development Goals [28]. To prepare the search strings, 15 sentinel papers were initially identified through simple searches in scientific databases using terms such as “trace elements” AND “beef cattle” and observing the relevance and scope of the papers retrieved according to the scope of interest (trace elements in beef cattle tissues in the last 22 years). The sentinel papers were used as a proxy to select only for the elaboration and verification of the search strings. Therefore, when applying the search string in scientific databases, sentinel papers should appear in the results; otherwise, it was an indication that the string should be improved. In this context, sentinel articles worked as a control to ensure the quality of the search engine in scientific databases. Search keywords were identified by analysing the titles and abstracts of the sentinel papers, and search strings were then created with the objective of retrieving all papers directly related to the sentinel papers and reducing the number of results outside the scope of this research. The details of the search performed, including the sentinel papers, can be seen in Appendix A.

A total of 3399 papers were obtained at the identification step, and 1507 duplicate papers were excluded. In the next step, the titles, abstracts and complete papers (when necessary) were examined, leading to the removal of 1542 papers considered outside the scope of this research and finally leaving a total of 350 papers.

At the document examination step, papers on the following topics were excluded: (1) research on dairy cattle; (2) research on young suckling calves; (3) research with emphasis on reproductive aspects; (4) research in which trace elements were only determined indirectly; and (5) research with emphasis on diseases resulting from deficiency or secondary toxicity by trace elements. Exceptions to exclusions were established for criteria 1, 2 and 4, when the environmental conditions were considered preponderant, e.g., clinical cases investigating trace element intoxication resulting from environmental contamination.

The papers accepted were subjected to an extraction phase to obtain general information such as year of publication, country where the research was conducted, authors and their affiliations and journal of publication. In addition, specific information was also obtained concerning the trace elements and bovine tissues analysed, analytical techniques and scope of the research with the One Health approach. Some papers that could not be accessed in full were analysed in relation to the information available in the databases, among which 15 papers were not available in digital format and another 8 papers required payment of a subscription.

Within the One Health approach, the papers were classified into animal health, environment health and human health, according to the aspects addressed in each. At first, we accepted all papers from this analysis in the animal health field in view of the search and selection prerequisites, which identified papers that analysed trace elements in bovine tissues. Although some papers did not directly address animal health, we felt that they all provided useful information on this topic. Within the topic of animal health, we highlighted those aspects related to animal handling, classified as animal category (age, sex and breed) and animal feeding (ration, forage and mineral supplementation). Environmental health was identified as a topic in papers that investigated natural (e.g., soil, water, climate and location) and anthropogenic (e.g., agriculture, mining and industry) aspects that can alter the nutritional status of animals. Finally, the human health topic was identified in papers that evaluated trace elements as nutrients or contaminants in human diets.

To proceed with the analysis of the co-authorship network, the correct spelling of the authors’ names was first verified and where necessary changed. A network map was then generated using VOSviewer software, version 1.6.16, which enables generation of network maps combining information from co-authorships to trace the graphical representation, in which the nodes represent the relevance of the authors and the links represent the papers published in co-authorship.

Bar charts, location maps and sector charts were generated using the ggplot2, hcmap and moonBook packages in R software, version 4.0.4. To assess whether the number of publications has tended to increase over the years, we performed a linear regression using the geom_smooth function, method lm from the ggplot2 package. In addition, we used the Inkscape software, version 1.0.1, for graphic adjustments and production of Boolean graphs.

## 3. Results

### 3.1. Overview of Papers by Years and Countries

The search for literature concerning trace elements in beef cattle production in the period studied (2000–2022) identified 350 papers that analysed trace elements in bovine tissue. Despite being an important research topic, the number of publications per year was quite variable (9 to 26 papers per year; average of 15.4 papers per year), with a moderate trend towards an increase in the number of papers on this topic published annually (Figure 1A), although the difference was not statistically significant. This upward trend reveals that the subject has received increasing attention from the scientific community in the period considered. It is therefore even more important to identify the main advances made and the main knowledge gaps, so that future actions are directed strategically and efficiently, providing opportunities to generate more accurate results specifically aligned with the various needs, whether in the local, regional or global context.

The searches also led to the retrieval of research carried out in 61 countries, covering all continents, as can be seen in the map in Figure 1B. This wide distribution demonstrates that the research topic is of global interest. The countries where the largest number of studies were conducted are also those where cattle breeding is of great importance [29,30], i.e., the US (83 papers), Spain (38 papers), India (32 papers) and Brazil (21 papers). Nonetheless, the research was concentrated in European countries and the US, indicating high levels of regional bias. 

Beef cattle are reared in many parts of the world, providing an important source of nutrients in the human diet [31], participating significantly in the economy of many countries [32] and representing an important expression of human socio-cultural diversity [33]. On the other hand, research investigating trace elements in animal tissue usually depends on the availability of researchers, analytical infrastructure and research incentives, among other aspects. Consistent with this perspective, the largest portion of research conducted in this period was carried out in European countries and the US, which are economically and scientifically developed, which may favor good conditions for conducting research on this topic.

### 3.2. Co-Authorship Network Analysis

Through the analysis of co-authorship network, it was possible to identify six main co-authorship networks in research and the main authors in each network (Figure 2). In this analysis, the nodes represent the authors, and the links represent the publication co-authorship. Co-authorship networks are important for scientific development in which scientists are encouraged to collaborate in order to make advances in knowledge, overcome growing specialization within science and share infrastructure in addition to combining different types of knowledge and skills to solve complex problems, thereby providing an opportunity to expand the scope of research and promote innovation [34]. The results obtained here allow us to visualize the networks with the greatest impact on the production of scientific knowledge and how these collaborations have been established, from which conclusions can be reached about the scientific advances achieved and the potential prospects.

Altogether, 1186 authors were identified. However, considering the visual aspect of the graph (Figure 2), only authors with more than one publication and only networks including authors with more than four publications are shown. Network 1 with 49 papers is mainly composed of researchers from the US, especially Jerry W. Spears, Stephanie L. Hansen and Terry E. Engle, affiliated respectively with the North Carolina State University, Iowa State University and Colorado State University. This group constitutes the largest collaboration network, with the largest number of researchers involved. In the sequence, Network 2 stands out, with 36 papers, represented mainly by researchers from Spain, affiliated to the University of Santiago de Compostela, especially Marta López-Alonso and Marta Miranda. This group includes the authors with the greatest number of publications within this research topic. Four other prominent networks are represented by research groups from Japan—Hokkaido University (Network 3–7 papers), Brazil—University of Sao Paulo (Network 4–5 papers), India—Odisha University of Agriculture & Technology and Indian Veterinary Research Institute (Network 5–8 papers), and Slovakia—Veterinary University Medicine in Kosice (Network 6–5 papers).

Among the networks identified, a distinction is made by mainly regional aspects, i.e., by the countries in which the networks were established. This stems from the regional aspects common to the authors, which may be favored by geographic proximity, language and regional research problems of related interests. In addition, the networks were also distinguished by the methodological approaches used and the general scope of the research. Network 1 is characterized by research related to animal health, addressing management aspects, mainly nutrition and animal performance, and the publications mainly appear in journals focused on Animal Science. The main research topics in this network include animal feed and monitoring of essential trace elements, particularly Cu and Zn, to assess animal performance. On the other hand, themes related to environment aspects and studies of potentially toxic elements, such as Pb and Cd, did not appear in this network. 

The other groups identified (Networks 2 to 6) are characterized by research with a broader and interdisciplinary scope. In these networks, the authors addressed environmental, human and animal issues within the research topics, e.g., differences in the location of the animal breeding areas in relation to the proximity of industrial or mining activities, evaluation of the risk of meat contamination and animal poisoning case studies in addition to practices adopted in animal production systems, such as feeding or animal performance. These networks study a greater variety of trace elements, and potentially toxic elements are often included. 

Furthermore, also in relation to analysis of the co-authorship network (Figure 2), we observed that the research groups identified (Network 1 to 6) included only 17% of all authors and 31% of all papers. On the other hand, the other networks included 83% of authors and 69% of the papers. This reveals that most authors involved and publications on this topic corresponded to emerging research groups. Therefore, it is possible that the expansion of collaborations among researchers could strengthen emerging groups and favor innovation in consolidated groups.

### 3.3. Trace Elements, Bovine Tissues and Analytical Techniques Identified

In total, 56 trace elements were studied in the papers reviewed, although only 13 of these elements were studied in more than 32 papers (Figure 3A). Among the most studied elements, Cu and Zn stand out, as analysis of these elements was reported in 194 and 164 papers, respectively. The third most studied element was Pb, considered in 117 papers. The other elements that received great interest in the research were Fe, Se, Cd, Mn, As, Co, Mo, Ni, Cr and Hg. 

The elements Cu, Zn, Fe, Se, Mn, Mo, Co, Ni and Cr are considered essential for ruminants and are therefore required for various biochemical and physiological functions of animals. These elements usually need to be added to animal feed to supply nutritional requirements [35,36]. Recent advances in ruminant nutrition have led to establishment of Cr and Ni requirements. Cr is considered essential because it increases insulin sensitivity, and Ni is essential for bacteria that exert an effect on the rumen microbial ecosystem, increasing the activity of the enzyme urease, thus favoring animal performance; however, there is some controversy about how essential these elements actually are for ruminants [37]. Regarding Mo, the dietary needs for cattle are not well defined because in practical feeding conditions, no deficiencies were observed [36].

The large number of studies investigating Cu and Zn in bovine tissue, as shown in Figure 3A, are explained by the generally low bioavailability of these elements in ruminant nutrition [38], and deficiencies are common worldwide [39]. Special attention has been given to Cu, which has been particularly challenging in bovine production. There is a delicate balance between Cu deficiency and toxicity, and antagonistic interactions occur with various other elements, such as Mo, S and Fe, resulting in numerous cases of Cu poisoning, especially in intensive production systems [40].

The elements Pb, Cd, As and Hg do not have established biological functions and are considered undesirable and potentially toxic contaminants of animal feed [3] in addition to constituting a danger to public health, due to the high degree of toxicity, even at very low levels of exposure [41]. These elements have generally been studied in relation to maximum tolerable levels, or safety levels, so that they do not harm animal or human health [42], mainly due to the potential for bioaccumulation along the food chain [43]. Excess levels of these elements in cattle are generally derived from anthropogenic emissions and food contamination [44]. However, as occurs with As in India, excess concentrations can also be of natural origin but be potentiated by anthropogenic actions [45]. 

Among the less studied trace elements (Figure 3B), 20 elements were included in between 2 and 13 papers, while 23 elements (others) were each included in only 1 paper. In general, of these elements (Figure 3B), only I is considered essential for ruminants. B, Li, Rb, Si and V have occasional benefits, and Al and F are potentially toxic [8]. 

The importance of trace elements is continuously discussed and revised, with the elements being classified as toxic, beneficial or essential for living organisms, especially humans [46,47]. It has been suggested that some elements such as, B, Rb, Si, V, F and Sn may be required by ruminants [42]. However, further information on the use of these and other elements is required to understand how they are distributed and interact in the organism at different stages of animal development [8]. Furthermore, it is important to determine the dynamics of these elements in agricultural systems [26] and in the environment in general, as well as the possible impacts of fluxes in the element concentrations [7].

Determining the nutritional requirements of trace elements is also important for identifying nutritional status and the subclinical detection of deficient or toxic concentrations through chemical analysis of animal tissue [48]. This can facilitate management in production systems and prevent pathologies resulting from the imbalance of these elements, as well as preventing excess concentrations of those elements that do not occur naturally in food consumed by humans [4].

Considering the bovine tissues analysed, liver, muscle and kidney predominated in the studies reviewed (Figure 3C). The liver is the main organ of animal metabolism responsible for storage of most trace elements and is therefore fairly representative of nutritional status [49]. It is also easy to sample [50], making it the main tissue analysed for determining trace elements in beef cattle (207 papers). The kidney, which is also an important organ of animal metabolism, was analysed in 101 papers, sometimes showing a distinct tendency to accumulate certain elements, being the main organ of accumulation of Cd [13] and also accumulating higher levels of other elements such as Pb [25] and Se [51].

Muscle was analysed in 128 papers and was generally found to have lower levels of trace elements than liver or kidney. On the other hand, muscle is the most commonly consumed by humans, and it assumes great importance in relation to meeting dietary requirements of essential nutrients and examining the risk of exposure to contaminating elements [52], and it is therefore frequently evaluated with a view to human food security.

Analysis of blood tissues and hair is very useful to assess bovine nutritional status in vivo, and it was also very representative in the papers reviewed, being included in 78 (blood plasma), 66 (whole blood), 62 (blood serum) and 23 (hair) papers (Figure 3C). Regarding blood analysis, we did not find any studies reporting differences in trace elements between whole blood and blood plasma in cattle, although human studies [53] have reported wide variations between these sample matrices for various elements. Regarding blood plasma and blood serum in cattle, Luna et al. [54] demonstrated that these matrices are equally suitable for determining several elements, except Cu and Se, which occurred at lower concentrations in serum than in plasma. These authors also suggested further research on Se should be conducted, as it is already known that part of the Cu in ruminant blood samples is sequestered during coagulation.

Analysis of bovine hair was proposed by Combs [55] as a potential method of verifying the mineral status of cattle, when combined with other indicators to increase the accuracy of assessment, as non-dietary factors such as sex, age, hair color, genetics, place of collection and sample contamination. Among the different types of non-invasive sampling, the use of hair must be improved to produce more accurate results. We have identified some advances such as a recommended sample preparation method for the chemical analysis of hair by the combined use of ethanol and ultrasound [56] and a recommended hair sampling location, on the withers [57]. Other studies have shown correlations between some elements in the blood and hair [58] in addition to confirming differences in relation to breed [59] and age of cattle [60].

Other bovine tissues have mainly been studied to assess compliance with dietary requirements and to identify patterns of metal accumulation due to breed, food and environmental conditions. These data are scarce, and discrepancies may therefore occur, as verified by Berata et al. [61], who observed greater accumulation of Pb in the lungs than in the liver in cows exposed to Pb, demonstrating the need to conduct further studies and thus improve the understanding of environmental or managerial aspects associated with animal metabolism involving different elements.

In terms of analytical techniques for trace element determination in bovine tissue, 30 different techniques were identified. The main techniques used were flame atomic absorption spectrometry (F-AAS) or graphite furnace atomic absorption spectrometry (GF-AAS) and multi-element techniques such as inductively coupled plasma mass spectrometry (ICP-MS) and inductively coupled plasma optical emission spectrometry (ICP-OES) (Figure 3D). In addition to the techniques already mentioned, we also highlight those generally used to determine specific elements, such as hydride generation atomic absorption spectrometry (AAS-HG) for As and Se determination, the fluorometric method for Se determination and biomarkers such as Ceruloplasmin and Glutathione peroxidase (GSH-Px) used for indirect determination of respectively Cu and Se.

Atomic absorption spectrometry (AAS) techniques are traditionally widely used. However, for analysis of complex matrices that require analysis of multiple elements, multi-element techniques such as ICP are more suitable, especially because they have enabled analysis of a wide range of elements including ultra-trace elements in the order of parts per trillion in addition to not requiring long sample pre-treatment steps, which is one of the main disadvantages of AAS [62].

ICP-OES provides an excellent, highly sensitive means of determining trace elements and can detect many elements simultaneously. However, the detection limit is similar to that of F-AAS. On the other hand, ICP-MS provides the best characteristics of all atomic spectrometry in terms of sensitivity, detection limits, transfer rate and multi-element measurement, although it is very expensive, despite having been developed over several years [63,64].

Other multi-elementary techniques of great potential for trace element analysis in bovine tissues are techniques that employ X-ray or neutron activation. In the present review study, we identified only four papers in which neutron activation analysis was used and two papers that used X-ray, indicating that these techniques can still be regarded as incipient for this type of study. We also observed that 40 of the trace elements identified in this research were analysed exclusively by multi-element techniques (mainly ICP-MS), which have led to improvements in the understanding of the dynamics of a wide range of trace and ultra-trace elements in the context of beef cattle production.

### 3.4. Scope of Papers in Relation to the One Health Approach

Identification of studies using the One Health approach followed three steps. First, we initially admitted 350 papers in this review within the animal health interface, considering that animal health was supported in the stages of search, identification and selection of papers. Next, we identified 201 papers in the field of environmental health. Finally, we identified 164 papers that addressed issues related to human health. The multiple issues involved in this research indicated the complexity of the challenges involved in animal health, conservation of the environment and safe production of food, which go beyond the limits of farming and are established on a large scale. Such issues often interact with each other and thus can be elucidated through the One Health approach (Figure 4A).

Interactions between the research topics occurred in 34 papers in the animal–human health interface, 71 papers in the animal–environment health interface and 130 papers in the One Health interface (animal–human–environment health). This demonstrates how closely the investigative approach to trace elements permeates natural aspects, human activity, animal feed, human nutrition and differences between animals (among others) in different ways.

We also observed that only 7% of the research studies carried out in the US were categorized as concerning the One Health interface (6 out of 83 papers), while 78% exclusively addressed the animal health interface (65 out of 83 papers). In comparison, 46% of surveys carried out in other countries (124 out of 267 papers) involved the One Health interface. This suggests that there is a distinction between the main research objectives that may be related to the interdisciplinary approach. In this case, in the US, research focused on animal health predominated, evaluating performance and productivity, while few studies integrated environmental and human health.

The papers analysed came from 157 journals and cover a broad scientific scope, with emphasis on journals in the fields of animal and veterinary science (Figure 4B). We also found a predominance of publications concerning animal health among some journals within the field of animal science, such as the *Journal of Animal Science*, the *Professional Animal Scientist* and the *Animal Feed Science and Technology*. On the other hand, the One Health approach predominated among interdisciplinary journals such as *The Science of The Total Environment* and *Food Additives & Contaminants—Part A*. This indicates that the main journals in which these studies are published also express the differences between the research approaches adopted.

Although the concept of One Health permeated just over one-third of the papers analysed (37%), we highlight its potential to be incorporated in future research on trace elements in beef cattle and in animal production in general, thus favoring new approaches to addressing emerging issues in the global scenario. Examples of issues related to trace element monitoring that could be enhanced by this approach include identification of exposure risk sites, health risk diagnosis, support for regulatory compliance and identification of safer and more effective agricultural practices, among many others [65]. Taking into consideration the studies reviewed here and to better elucidate our point, we suggest expanding the factors investigated (soil, feed, animal physiology, nearby anthropogenic activities) and the number of trace elements included in each search (essential and toxic elements). In this context, we recommend reading a framework for One Health research [66], which offers guidance to researchers regarding the practical design and implementation of One Health research. In addition, we also recommend some papers in which trace elements are considered within the animal–environment–human health interface: [12,13,25] (others in Appendix A).

### 3.5. Gaps and Opportunities

The information presented here refers to the number of papers in relation to year of publication, country where the research was carried out, authors involved, trace elements studied, bovine tissues analysed, analytical techniques used and the field of research. Both qualitative and quantitative information was reported, thus demonstrating the scientific progress made in this area. However, some knowledge gaps were identified: (1) lack of research in developing countries; (2) need for answers to a wide variety of questions related to the environment and the conditions under which livestock is raised; and (3) need to address the functions and interactions of a wide range of trace elements in animal metabolism and their relationship to environmental and human health.

Few changes have been made in the trace mineral requirements of beef cattle since the sixth edition of the National Research Council (NRC) in 1984 [35]. However, we have identified a high potential for scientific advances in this area of knowledge with the possible expansion of information on the management of trace elements in livestock, which may be favored by incorporation of the One Health approach in future research.

Despite the greater complexity of the One Health approach, which involves multiple issues, this approach has become widely accepted. Research with inter-, multi- or transdisciplinary approaches has been necessary to provide greater integration between disciplines and experts, as well as a broader interpretation or application of the results obtained [67]. In this context, the Farm to Fork strategy stands out, an integral part of the European Commission’s Green Deal, which emphasizes the need for integrated approaches to advance towards safe and sustainable food systems to improve human, animal and environmental health simultaneously [68].

There are complex interactions that occur in the production system, which combine environmental conditions in addition to social, economic and cultural dimensions, resulting in unique characteristics in each location, which need to be recognized in research planning. These interactions require specific approaches to obtaining and interpreting results. In view of this, there should be greater interaction and collaboration among researchers to develop this type of research, which should favor greater sharing and advancement of knowledge.

In the One Health approach, knowledge of the basic disciplines is reinforced, as this approach plays a fundamental role in aggregating complementary disciplines and different areas of expertise [66,67,68,69,70]. The structure of the new research can thus have a wider scope and integrate broader issues, without omitting the characteristics of the basic areas of knowledge. Furthermore, collaborations can help to optimize research resources and infrastructure, to overcome regionalization and ensure safe production of food in the long-term throughout the world.

Advances in the development and improvement of analytical techniques for the knowledge of the content of trace elements in bovine tissue have been shown to be essential. However, it is also important to consider access to cutting-edge technologies in less developed countries, especially due to the associated costs, which may constitute an important barrier limiting research in these regions. For instance, in Africa, the collaboration between national and international research centers provide better approaches to promote One Health technologies in remote areas centers [71].

Regarding the 17 Sustainable Development Goals (SDGs) [28], it is essential to understand livestock in a holistic perspective, so that it is possible to lead animal production to contribute to all these established objectives [72]. Therefore, scientific production in livestock should also reflect this perspective in all its fields of knowledge. This could contribute to efficient production models adapted to different socioeconomic and environmental contexts, more balanced and efficient supplements, more rational use of natural resources, less impact on climate change and production of healthier foods, for instance.

New research should observe in an increasingly integrated way the environmental aspects of livestock and food quality to enable the advance towards precision livestock and reach Sustainable Development Goals. Thus, trace elements cannot be investigated in isolation within a production system, and integration of data that combines information from micro and macro systems, covering sociocultural, economic and especially environmental dimensions is required.

## 4. Conclusions

This review enabled us to summarize the information obtained between 2000 and 2021 about trace elements in the production of beef cattle and to identify the main advances and knowledge gaps. The main research groups are located in the US and Spain, although most publications (69%) were produced by emerging research groups investigating this topic. Among the studies carried out in the US, only 7% were categorized as the One Health approach, while in other countries the corresponding proportion was 46%. Overall, only 37% of studies addressed aspects that integrated animal, environmental and human health.

We found that 56 trace elements and 15 bovine tissues were studied; however, the main research targets were the elements Cu, Zn and Pb and the bovine tissues liver, muscle and kidney. We also noted the importance of improving analytical techniques to increase the number of trace elements studied and improve understanding about these elements in different contexts of animal production.

Finally, we highlight the potential for extending international scientific cooperation to promote the inclusion of developing countries and greater collaboration across research disciplines in order to produce more significant advances in knowledge and more innovative and accurate responses to emerging issues worldwide. In this context, we suggest adoption of the One Health approach in planning new research on trace elements in animal production.

## Figures and Tables

**Figure 1 animals-12-02254-f001:**
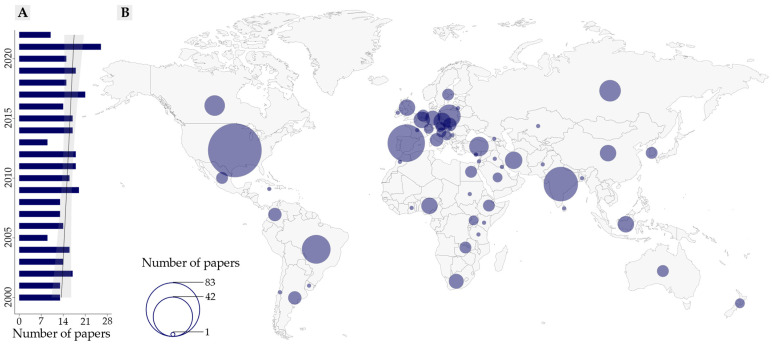
Number of papers concerning trace elements in beef cattle production published in the period 2000–2022: (**A**) bar plot and trend line, with confidence interval (shaded area) for papers published by year, between 2000 and 2022; data referring to 2022 were obtained up to 1 July 2022 and adjusted to the 12-month period by multiplying by 2; (**B**) map showing the number of papers published in different countries.

**Figure 2 animals-12-02254-f002:**
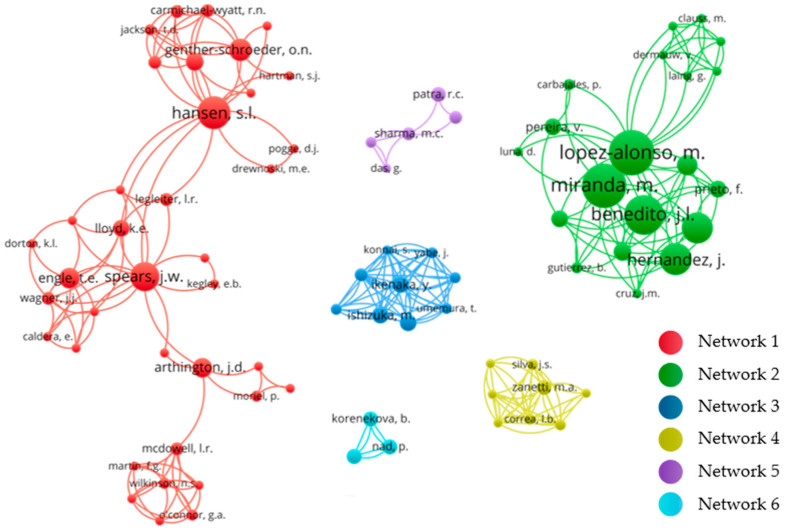
Co-authorship networks active in the period 2000–2022 in research on trace elements in beef cattle production. The nodes represent the relevance of the authors; the links represent the co-authorship among authors, and the colors represent the main networks identified.

**Figure 3 animals-12-02254-f003:**
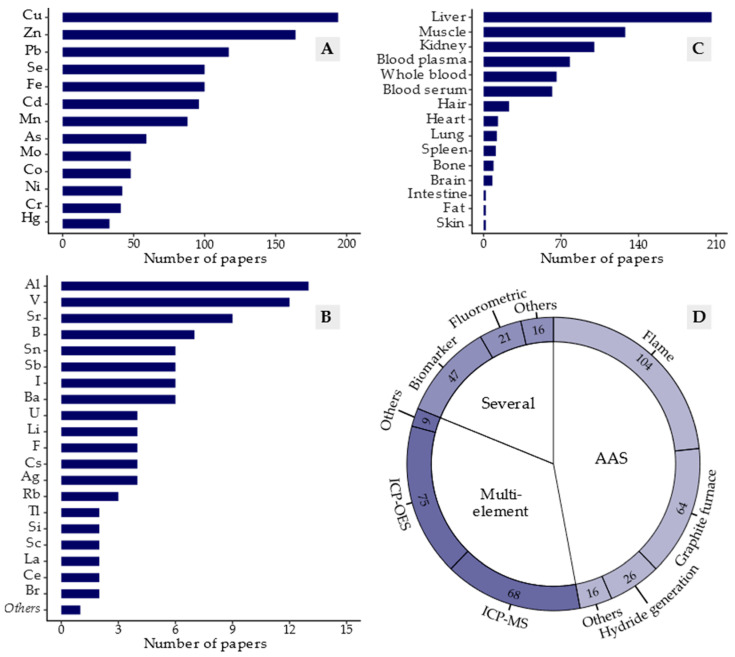
Number of papers published in the period 2000–2022 about trace elements in beef cattle production: (**A**) trace elements most studied; (**B**) trace elements less studied; Others elements: Ar, Au, Be, Bi, Dy, Er, Eu, Ga, Gd, Hf, Ho, In, Ir, Lu, Nd, Pr, Pt, Sm, Te, Ti, Th, Tm, Yb; (**C**) bovine tissues analysed; (**D**) main analytical techniques used.

**Figure 4 animals-12-02254-f004:**
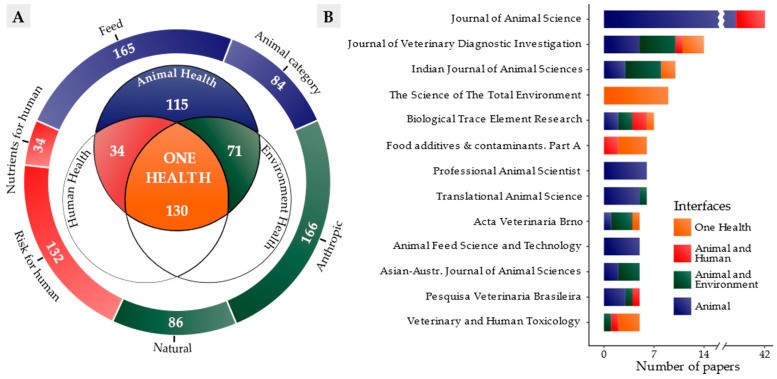
Number of papers published in the period 2000–2022 about trace elements in beef cattle production classified by research scope within the One Health approach: (**A**) interface of number of papers categorized by research aspects and research scope; (**B**) number of papers categorized by main journals and by research scope.

## Data Availability

Not applicable.

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
