# Peer review of "Trace Elements in Beef Cattle: A Review of the Scientific Approach from One Health Perspective"

_animals, 2022, doi:10.3390/ani12172254_

Round 1
Reviewer 1 Report
According to ,,livestok'' definition this therm include also other types of farm animals, not just cattle. Study was focus on beef cattle published papers. So, please adapt the title of your manuscript according to animal species that you studied.
Please explain how you selected the reference list (that contains only 70 titles) from those 350 papers identified by you for this study.
In my opinion this review is particular. It has a specific point of view, by considering the number of publications per year, 24 country, authorship. I think that these parts are not relevant for the aim of the study, presented by the authors in the ,,Abstract'' and ,,Introduction'' and should be eliminated from this study, not being relevant for the One Health approach.
I think that the other parts ( trace elements, bovine tissues, analytical techniques, and scope of the papers) are more relevant, by considering this study from the ONE HEALTH perspective.
I consider that only these parts ( trace elements, bovine tissues, analytical techniques, and scope of the papers) are relevant and should be keept in this study. So, I recommend to restructure the review and to detail more the most important subjects from the One Health point of view.
Author Response
Dear reviewer,
Thank you very much for your time and effort in reviewing our ms. We are very grateful for your positive comments which have been considered in the new version. The ms has substantially improved, the objective being much clearer for the reader. Hope you find it now appropriate for publication in Animals.
Reviewer 1 Report
Comments and Suggestions for Authors
According to ,,livestok'' definition this therm include also other types of farm animals, not just cattle. Study was focus on beef cattle published papers. So, please adapt the title of your manuscript according to animal species that you studied.
ANSWER: We accept the suggestion made by the reviewer and we have modified the title from 'livestock' to 'beef cattle'. Now the title is – “Trace elements in beef cattle: A review of scientific approach from One Health perspective”
Please explain how you selected the reference list (that contains only 70 titles) from those 350 papers identified by you for this study.
ANSWER: We understand the comment made by the reviewer and realised that the objective of the paper was not well described in the ms. Therefore, we have better stated that in this work we did not intend to discuss the findings of those 350 papers, but rather to analyse the general approach adopted by them. In this way, we understand that there is no need to cite them directly in the text or even for them to be present in the references. On the other hand, all papers can be consulted in the supplementary files. Among these 350 papers, only those that we considered relevant to elucidate specific arguments were cited.
In my opinion this review is particular. It has a specific point of view, by considering the number of publications per year, 24 country, authorship. I think that these parts are not relevant for the aim of the study, presented by the authors in the ,,Abstract'' and ,,Introduction'' and should be eliminated from this study, not being relevant for the One Health approach.
I think that the other parts ( trace elements, bovine tissues, analytical techniques, and scope of the papers) are more relevant, by considering this study from the ONE HEALTH perspective.
I consider that only these parts ( trace elements, bovine tissues, analytical techniques, and scope of the papers) are relevant and should be keept in this study. So, I recommend to restructure the review and to detail more the most important subjects from the One Health point of view.
ANSWER: We totally agree with the reviewer and we understand that our writing needed to be clearer. In our opinion the information regarding years, countries and authorship is essential to understand the context in which this research was carried out. In fact, this is the approach followed by other review papers following a systematic scientiometric review methodology. We hope that the adjustments made to the title, abstract and objectives are adequate to allow this clarification. We have now changed parts of the text such as: “The aim of this review was to collect and globally summarize the research carried out in the last few decades concerning trace elements in beef cattle and analyse the approach of researchers through bibliometric information (years, countries, authorship and journals) and topics (trace elements, bovine tissues, analytical techniques and animal, environment and human health) presented in their respective papers” in the aim of the study in the introduction section.

Reviewer 2 Report
The article is well written and put together, however, title is misleading. There was a large amount of information on the number of articles that fit the One Health perspective, but no detailed information as to how those articles fit one health or the actual list of knowledge gaps or advances. Additionally, the sustainability goals are mentioned, but never discussed. Suggest adding more information in the discussion of detailed knowledge gaps, the information provided is extremely vague and leads into the blanket statement of "we need more research". What technical information was considered, how can it be utilized in the industry, and what additional technical information is needed?
Line 76: in addition, interacting...
Line 125: relevance and scope related to what?
How were the sentinel papers selected? They are mentioned several times and then in the discussion are never discussed again or how they fit into the models presented? Add citations for the 15 sentinel papers after they are first mentioned. There never was a clear review of the actual work, only a review of the number of papers that fit their definition.
Author Response

(The authors gave the same response as above.)

Reviewer 2 Report
Comments and Suggestions for Authors
The article is well written and put together, however, title is misleading.
ANSWER: We really agree to adjust the title. Thus, it we modified 'scientific advance' to 'scientific approach'. We also made improvements in the summaries and objective to clarify our idea. We emphasize that we did not intend to discuss the findings of those 350 papers, but rather to analyse the general approach adopted in these studies. The new title is now: Trace elements in beef cattle: A review of scientific approach from One Health perspective
There was a large amount of information on the number of articles that fit the One Health perspective, but no detailed information as to how those articles fit one health or the actual list of knowledge gaps or advances.
ANSWER: We are sorry if we have not understood properly the reviewer comment. Regarding “but no detailed information as to how those articles fit one health”, the explanation about how the papers were classified is presented in the Material and methods section (lines 234-245) and in section 3.4. (lines 485-494).
Regarding “but no detailed information as … the actual list of knowledge gaps or advances”, the gaps or advances were identified based on the frequency of topics in each paper.
Additionally, the sustainability goals are mentioned, but never discussed. Suggest adding more information in the discussion of detailed knowledge gaps, the information provided is extremely vague and leads into the blanket statement of "we need more research".
ANSWER: We appreciate the comment and have added some discussion regarding sustainability goals and adjusted our statements. The text now is: “Regarding the 17 Sustainable Development Goals (SDGs) [28], it is essential to understand livestock in a holistic perspective, so that it is possible to lead animal production to contribute to all these established objectives [72]. Therefore, scientific production in livestock should also reflect this perspective in all its fields of knowledge. This could contribute to efficient production models adapted to different socioeconomic and environmental contexts, more balanced and efficient supplements, more rational use of natural resources, less impact on climate change and production of healthier foods, for instance”.
What technical information was considered, how can it be utilized in the industry, and what additional technical information is needed?
ANSWER: When we propose to evaluate the approach adopted in the research works, we assume bibliometric information and information related to the frequency of occurrence of topics of interest. We hope that the adjustments we have made to the text will allow this clarification. “The aim of this review was to collect and globally summarize the research carried out in the last few decades concerning trace elements in beef cattle and analyse the approach of researchers through bibliometric information (years, countries, authorship and journals) and topics (trace elements, bovine tissues, analytical techniques and animal, environment and human health) presented in their respective papers”.
Line 76: in addition, interacting...
ANSWER: We have now modified this wording.
Line 125: relevance and scope related to what?
ANSWER: Regarding the scope of search interest. We have included this information in the text: “To prepare the search strings, 15 sentinel papers were initially identified through simple searches in scientific databases using terms such as "trace elements" AND "beef cattle" and observing the relevance and scope of the papers retrieved according to the scope of interest (trace elements in beef cattle tissues in the last 22 years)”.
How were the sentinel papers selected? They are mentioned several times and then in the discussion are never discussed again or how they fit into the models presented? Add citations for the 15 sentinel papers after they are first mentioned.
ANSWER: We accept the point made by the reviewer and we have added this information in the ms: “The sentinel papers were used as a proxy to select only for the elaboration and verification of the search strings. Therefore, when applying the search string in scientific databases, sentinel papers should appear in the results, otherwise, it was an indication that the string should be improved. In this context, sentinel articles worked as a control to ensure the quality of the search engine in scientific databases”. The sentinel papers are described in the supplementary file A.
There never was a clear review of the actual work, only a review of the number of papers that fit their definition.
ANSWER: We understand that due to the lack of clarity in the wording of the title, abstracts and objectives, we may have generated an expectation that the focus of the review would be to discuss the main findings of each paper. We hope that the adjustments made to the title, abstracts and objectives are adequate to allow this clarification. We have substantially changed the text and believe that now we present a clear review of the actual work, such as: “The aim of this review was to collect and globally summarize the research carried out in the last few decades concerning trace elements in beef cattle and analyse the approach of researchers through bibliometric information (years, countries, authorship and journals) and topics (trace elements, bovine tissues, analytical techniques and animal, environment and human health) presented in their respective papers.”

Reviewer 3 Report
Review of Manuscript Animals-1843585
The paper reviewed trace elements in beef cattle from the One Health perspective. In general, the paper gives a very good overview about research on trace elements done during the last 2 decades. The paper is very well written, with a good structure and comprehensible. The paper gives a good overview about literature available in the topic, advances done during the last years, highlighted some gaps about research topics, identified opportunities. The manuscript is a good guidance and update for researchers dealing with trace minerals. I have no comments for the authors about the paper
Author Response
Thank you very much for your time and effort in reviewing our ms. We are very grateful for your positive comments.

Round 2
Reviewer 1 Report
The authors corrected the manuscript according to reviewer recommendations.